# Collation Efficiency of Poly(Vinyl Alcohol) and Alginate Membranes with Iron-Based Magnetic Organic/Inorganic Fillers in Pervaporative Dehydration of Ethanol

**DOI:** 10.3390/ma13184152

**Published:** 2020-09-18

**Authors:** Gabriela Dudek, Roman Turczyn, David Djurado

**Affiliations:** 1Department of Physical Chemistry and Technology of Polymers, Faculty of Chemistry, Silesian University of Technology, Strzody 9, 44-100 Gliwice, Poland; roman.turczyn@polsl.pl; 2Systèmes Moléculaires et nanoMatériaux pour l’Energie et la Santé (SyMMES), IRIG, CNRS, CEA Grenoble, Université Grenoble Alpes, 17 rue des Martyrs, 38054 Grenoble, France; david.djurado@cea.fr

**Keywords:** poly(vinyl alcohol), sodium alginate, iron oxides, iron(III) acetylacetonate, pervaporation, ethanol dehydration

## Abstract

Hybrid poly(vinyl alcohol) and alginate membranes were investigated in the process of ethanol dehydration by pervaporation. As a filler, three types of particles containing iron element, i.e., hematite, magnetite, and iron(III) acetyloacetonate were used. The parameters describing transport properties and effectiveness of investigated membranes were evaluated. Additionally, the physico-chemical properties of the resulting membranes were studied. The influence of polymer matrix, choice of iron particles and their content in terms of effectiveness of membranes in the process of ethanol dehydration were considered. The results showed that hybrid alginate membranes were characterized by a better separation factor, while poly(vinyl alcohol) membranes by a better flux. The best parameters were obtained for membranes filled with 7 wt% of iron(III) acetyloacetonate. The separation factor and pervaporative separation index were equal to 19.69 and 15,998 *g⋅m^−2^⋅h^−1^* for alginate membrane and 11.75 and 14,878 *g⋅m^−2^⋅h^−1^* for poly(vinyl alcohol) membrane, respectively.

## 1. Introduction

Commonly applied in laboratory practice, iron oxides possess numerous physicochemical properties responsible for their broad range of applications in various fields of science, including biology, geochemistry, mineralogy, and materials engineering [1]. Particularly interesting are magnetic properties of iron oxides, affecting their different interaction with the magnetic field. Elementary particles, besides the commonly known characteristics such as mass or charge, have also an internal angular momentum called a spin, resulting from the rotation of a particle around its own axis. In quantum mechanics, it corresponds to the so-called spin quantum number **s**. Particles with a non-zero spin also have a non-zero magnetic angular momentum, and one of the examples of such particles are iron oxides. Among many possible applications, magnetic properties of iron oxides can be used for the separation of molecules having similar physical properties, for instance in the pervaporative separation of water and ethanol.

Pervaporation, as a potential replacement for distillation process in several applications, has made a considerable progress in recent decades. In general, the pervaporation process can be divided into the hydrophilic and organophilic processes due to the nature of separated mixtures. The application of organophilic pervaporation involves a separation of organic low-concentrated wastes from organic-water mixtures, i.e., a removal of organic substances from ground or drinking water, or a separation of organic substances from a mixture of organic compounds, including a separation of benzene from benzene/cyclohexane mixture. The application of hydrophilic pervaporation refers to the removal of water from organic liquids, such as a dehydration process of ethanol or isopropanol [2,3,4,5,6]. Furthermore, the pervaporation process has also found successful application in assisting chemical and biochemical reactions [7], recovery of aromatic compounds from the food systems [8], and production of non-alcoholic beverages [9].

The development of pervaporation membranes for dehydration of organic solvents was mainly based on the use of hydrophilic polymers, such as poly(vinyl alcohol) (PVA), cellulose derivatives, chitosan, and alginate (ALG). However, these membranes are susceptible to excessive swelling, therefore, membrane stabilization processes are usually required. One of the strategies applied to strengthen hydrophilic membranes is to combine the inorganic filler with a polymeric matrix [10,11,12,13]. Such hybrid membranes possess some distinct and unique advantages compared with plain polymeric and inorganic membranes, i.e., high modifiability, enhanced selectivity and permeability. Therefore, different combinations of polymers and inorganic materials have been intensively studied in recent years. The most commonly used inorganic fillers are zeolites, MWCNT, silica, graphene oxides, metal particles, and metal oxides [14,15,16,17,18,19].

Problems in ethanol dehydration result from the similarity of water and ethanol molecules’ physical properties, especially polarity, making the membrane separation process challenging. The different interaction of water and ethanol molecules with magnetic field could be proposed to support the separation of these two components. Each atom, besides the commonly-known characteristics such as mass or charge, has also an internal angular momentum called spin which results from the rotation of particle around its own axis. In quantum mechanics it corresponds to the so-called spin quantum number **s**. Particles with a non-zero spin also have a non-zero magnetic angular momentum. The separation properties of membranes can be improved through the practical applications of the fact that water and ethanol differ in polarity and dipole moment. Water is a dipole, so it interacts strongly with magnetic substances, whereas ethanol is less affected by the magnetic field due to its lower polarity. Taking into account these difference in the physical properties of water and ethanol molecules, our recent research has been focused on application of three species containing iron, i.e., organic complex tris(acetylacetonato) iron(III), and inorganic oxides—hematite (Fe(III)_2_O_3_ and magnetite Fe(II,III)_3_O_4_) as fillers in PVA and ALG membranes. Hematite is regarded as the most stable iron oxide and environmentally friendly semiconductor. α-Fe_2_O_3_ has a rhombohedral structure containing the space group R-3c with n-type semiconducting properties. Depending on the morphology, crystallinity, and inter-particle interactions between hematite particles, α-Fe_2_O_3_ can display antiferromagnetic, weak-ferromagnetic or superparamagnetic properties [20]. Magnetite is a black powder serving as an interesting type of magnetic materials, and is broadly used in the fields of nanoscience and nanotechnology, information storage, magnetic fluids, and medicine. It has octahedral structure with an integral spin moment per formula unit equal to 4.0 μ_B_. Its minority spin electrons are conducting, whereas the majority-spin ones are insulating [21]. Tris(acetylacetonato) iron(III) (Fe(acac)_3_) is a red, air-stable ferric coordination complex featuring acetylacetonate (acac) ligands, soluble in nonpolar organic solvents. It has octahedral structure with six-fold coordination of central atom by oxygen atoms. Due to the occurrence of five unpaired d-electrons, Fe(acac)_3_ is paramagnetic, with a magnetic moment of 5.90 μ_B_ [22].

All proposed fillers have already been investigated in various fields, also in different aspects of membrane processes, e.g., hematite was applied for a removal of natural organic matter [23], as membrane filler in solid oxide fuel cells [24], as a catalyst in reactive membranes [25], ultrafiltration [26] and removal of heavy metals [27]. Magnetite was applied as membrane filler for pervaporation and membrane distillation [28], microfiltration [29], and a removal of heavy metals [30]. Tris(acetylacetonato) iron(III) has been widely used in organic synthesis, as well as in membrane preparation, e.g., as micropore formatting agent for pervaporative dehydration of isopropanol and gas separation membranes [22,31]. Notwithstanding, these recognized iron-based compounds with different nature and magnetic character have not been compared so far in terms of their efficiency as fillers in hybrid membranes intended for a pervaporative dehydration process. The use of these iron compounds as fillers in two series of hydrophilic membranes, one based on PVA and the other based on ALG matrix, allowed us to evaluate and compare the transport behavior and separation properties in the pervaporation process of water/ethanol separation. The influence of different iron compounds, their loadings and matrix nature were discussed and compared with plain membranes. Additionally, the physico-chemical properties of the resulting membranes were studied by microRaman spectroscopy, scanning electron microscopy, X-ray diffraction, contact angle and swelling measurements, to provide extensive characteristics of fabricated materials.

## 2. Experimental

### 2.1. Materials

Poly(vinyl alcohol) (PVA), sodium alginate (ALG), glutaricdialdehyde (2.5 wt.% solution in water), sodium hydroxide (purity ≥ 98%), sufuric acid (purity ≥ 95%), 2,2′-(ethylenedioxy)-bis(ethylamine) (purity ≥ 98%), iron(III) chloride, iron(II) chloride tetrahydrate, 2,4-pentanedione (+99%), ethanol (96%, extra pure), and methanol (99.8%, ACS reagent) were obtained from Acros Organic (Geel, Belgium). Calcium chloride (purity ≥ 96%), ammonia solution (25 wt.% solution in water), silver nitrate were purchased from Avantor Performance Materials S.A. (Gliwice, Poland).

### 2.2. Preparation of Iron-Based Fillers

Magnetite and hematite were obtained by a precipitation method on the basis of the protocols described in [32] and [33], respectively. Shortly, FeCl_3_ in case of hematite, and a mixture of FeCl_2_ and FeCl_3_ in case of magnetite were dissolved in deionized water. Then, 2M ammonia solution was added dropwise to the respective solutions and further stirred for about 60 min. As-formed precipitates were washed with an appropriate mixture until the chloride ions were completely washed out. The resulting particles were filtered and dried at 40 °C. The obtained hematite and magnetite were grounded in a planetary ball mill to obtain particles of D50 ~650 nm. Fe(acac)_3_ was synthesized through an iron(III) hydroxide two step pathway according to Chaudhuri and Ghosh [34]. Briefly, anhydrous FeCl_3_ was dissolved in water and hydrolyzed by a slow addition of an excess of ammonia solution with constant stirring and gentle warming. Next, the mixture was heated to 80 °C for 15–20 min. As-formed precipitate of iron(III) hydroxide was filtered off and washed to remove the chloride. Freshly prepared, moist iron(III) hydroxide was mixed with an excess of pentane-2,4-dione (molar ratio 1:4.5) and heated to 80 °C for 35 min. Then, the mixture was left to cool down. Obtained crystals of Fe(acac)_3_ were filtered off, dried and purified by recrystallisation form a methanol–water mixture. The obtained red crystals were grounded in a planetary ball mill to obtain particles of D50 ~650 nm.

### 2.3. Membrane Fabrication

1.0 wt% PVA and 1.5 wt% ALG solution were obtained by dissolving an appropriate amount of PVA or sodium alginate powder in deionized water. These solutions were mixed with an appropriate portion of magnetite, hematite and Fe(acac)_3_, to give the filler concentrations of 0, 3, 5, 7, 10 and 15 wt%. Iron-containing PVA or ALG solutions were then casted onto a levelled glass plate and evaporated to dryness at 60 °C. After 24 h, the PVA membrane was crosslinked by immersion in 50 wt% glutaraldehyde solution for 15 min. Alginate membrane was crosslinked using calcium chloride by the immersion in 2.5 wt% calcium chloride solution for 120 min at room temperature. Plain PVA and ALG membranes were prepared in the same manner as above, except for the addition of investigated fillers. The thickness of membranes was measured using waterproof precise coating thickness gauge MG-401 (Elmetron, Zabrze, Poland), estimated as a mean value of at least 10 measurements in different points, and equal to 18.0 ± 2.0 μm in case of ALG and 22 ± 2.0 μm in the case of PVA membranes, respectively.

### 2.4. Physico-Chemical Characterization

Magnetite, hematite and Fe(acac)_3_, as well as plain and hybrid membranes were characterized using several physico-chemical techniques, including microRaman spectroscopy, X-ray diffraction (XRD) measurements and scanning electron microscopy (SEM). Raman spectra were measured using InVia microRaman spectrometer (Renishaw plc., New Mills, Gloucestershire, UK) with 633 nm excitation laser source and 3 mW power at sample. Powder samples of iron oxides and Fe(acac)_3_ were subjected to XRD analysis in the θ–2θ geometryusing a Philips X’Pert diffractometer (PANalytical B.V., Almelo, the Netherlands) with Cu K-alpha X-rays source (λ = 1.542 Å). The measurements were conducted on monocrystalline silicon plate in a reflection mode, and 2θ range between 15–100° for inorganic oxide and 5–50° for organic complex. Diffractograms were analyzed using HighScore Plus^®^ software packages (Malvern Panalytical B.V., Almelo, Netherlands) coupled with ICDD PDF-4+ database (ICDD, Newtown Square, PA, USA). SEM images were captured using Phenom Pro-X microscope (ThermoFisher Scientific Inc., Waltham, MA, USA) equipped with an energy dispersive X-ray analyzer. Basing on the sorption tests in water, the degree of swelling (DS) was calculated using a following equation:(1)$$DS=\frac{W_{wet}-W_{dry}}{W_{dry}}\cdot 100\;\left(\%\right)$$ where *W_wet_* is the weight of a wet membrane, and *W_dry_* is the weight of a dry membrane.

For sorption tests, the pieces of membrane were immersed in water and allowed to reach the equilibrium swelling for 48 h. During this period the changes in the weight of samples’ were determined using analytical balance. Contact angles of the dry membranes were measured using an optical contact angle measuring and contour analysis systems OCA 15 from DataPhysic (DataPhysics Instruments GmbH, Filderstadt, Germany). The contact angle was measured immediately after the dropping and at the intervals of 10 s. To measure the effect of membrane fouling, 1 μL of droplet volume of deionized water was used. Sorption tests and contact angle measurements were repeated ten times, and average values were provided.

### 2.5. Pervaporation Experiments

Pervaporation experiments (PV) were carried out using the apparatus described in a previous paper [27] under the same conditions. A single measurement takes ten hours. However, this measurement is repeated at least three times for the same membrane. As a feed solution, an aqueous solution of 96 vol% ethanol was used. The permeate was collected in a cold trap cooled with liquid nitrogen. Flux was calculated from the measured weight of liquid collected in the cold traps during a certain time interval at steady-state conditions. The ethanol content in feed, permeate and retentate was analyzed by gas chromatography using Clarus 500 chromatograph (PerkinElmer, Shelton, CT, USA) equipped with a 30 m long elite-WAX ETR column. For each membrane, the separation process was repeated three times. The results showed good repeatability with standard deviation (SD) less than 2%. The permeation flux of component *i* was calculated using the following equation [35,36]:(2)$$J_{i}=\frac{m_{i}}{At}$$ where *m_i_*—weight of component *i* in permeate, *g*, *A*—effective membrane area, *m^2^*, *t*—permeation time, *h*.

Two parameters were used for the description of the separation properties of the membrane, namely separation factor (*α_AB_*) and selectivity coefficient (Sc_AB_). Separation factor was calculated by [35,36]:(3)$$\alpha_{AB}=\frac{y_{A}/y_{B}}{x_{A}/x_{B}}$$ where *x_A_*, *x_B_*—weight fraction of components in the feed, *wt%*, *y_A_*, *y_B_*—weight fraction of components in permeate, *wt%*.

Based on the 1st Fick’s law, the permeation coefficient was determined according to the formula:(4)$$P=\frac{J_{s}\cdot l}{\Delta p}$$ where:


*P*—permeation coefficient, $Barrer=\frac{cm_{STP}^{3}\cdot cm}{cm^{2}\cdot s\cdot cm_{Hg}}\cdot 10^{10}$,

*l*—membrane thickness, *cm*,

*Δp*—difference of vapour pressure at both sides of the membrane, *cm_Hg_*.

*J_s_*—diffusive mass flux, $\frac{cm_{STP}^{3}}{cm^{2}\cdot s}.$

The diffusion coefficient was estimated using the method described in [37]. This model assumes that before measurement, the analyzed membranes are not empty. It also considers the tubing length between the permeation cell and the cold traps. According to this method, the diffusion coefficient was calculated by:(5)$$D=\frac{-l^{2}}{3L_{a}}$$where:


$L_{a}$—the effective total Time Lag calculated as $L_{a}=L_{a}^{2}-6.5L_{a}^{1}$;

$L_{a}^{1}$—time lag for the tubing, *s*

$L_{a}^{2}$—the asymptotic Time Lag, *s*

*l*—the thickness of membrane, *cm*

The solubility coefficient is a measure of the sorption in membrane. It is characterized by splitting of the penetrant between the membrane and the outer phase in equilibrium. This parameter was calculated from the relation:(6)$$S=\frac{P}{D}$$where:


*S*—solubility coefficient, $\left[\frac{cm_{STP}^{3}}{cm^{3}\cdot cm_{Hg}}\right].$

Selectivity coefficient is equal to the ratio of permeation coefficients of separated components [35,36]:(7)$$Sc_{AB}=\frac{P_{A}}{P_{B}}$$

In order to compare the separation efficiency of investigated membranes, pervaporation separation index *(PSI*) expressed by a following equation [35,36] was used:(8)$$PSI=J\left(\alpha_{AB}-1\right)$$ where *J* is the total permeate flux, *α_AB_* is the separation factor.

## 3. Results and Discussion

### 3.1. Membrane Characterization

#### 3.1.1. Raman Studies

Raman spectra of iron-based filler particles are shown in Figure 1. The spectrum of hematite particles showed characteristic six lines typical for α-Fe_2_O_3_ crystal at about 225, 292, 405, 490, 602, and 1305 cm^−1^. The lines at 225 and 292 cm^−1^ were associated with the movements of iron ions and the lines at 404 and 490 cm^−1^ were attributed to a symmetric breathing mode of the oxygen atoms relatively to each cation in the plane perpendicular to the crystallographic c axis. Apart from the symmetry-allowed modes, Raman spectra of hematite exhibited two extra lines at about 602 and 1305 cm^−1^, originating from tetrahedral defects [38]. The spectrum of magnetite, which has a spinel structure at room temperature and contains both Fe^2+^ and Fe^3+^ cations, possessed three visible Raman bands at about 306, 532, and 667 cm^−1^ assigned as symmetric bends of oxygen with respect to the iron atom, asymmetric stretch of iron and oxygen atoms, and a symmetric stretch of the oxygen atoms along the Fe–O bonds, respectively [39]. The two weakest bands at 193 and 410 cm^−1^ were not found in the spectrum. Fe(acac)_3_ exhibited absorption bands in the range of 170–1600 cm^−1^, which were described by a superposition of the ligand-to-metal charge-transfer excited states and the ligand (π, π*) excited triplets and singlets. The most intense Raman feature was observed at 452 cm^−1^, and was assigned to a symmetric stretch of the ligands relative to the iron atoms υ(Fe–O). O–Fe–O rocking motions were visible as bands at 172 and 218 cm^−1^. The broad band at 1605 cm^−1^ was associated with the stretching υ(C–O) of the (acac) ligands as well as their most pronounced skeletal vibration at 1280 and 1370 cm^−1^ [40].

Raman spectra of plain and hybrid PVA and ALG membranes filled with Fe(acac)_3_ particles are shown in Figure 2. Pristine PVA membrane exhibited several characteristic peaks of C–H different deformation modes, the strong one at 1443 cm^−1^ (shear mode), at 917 and 857 cm^−1^(fan or twist mode) from the syndio- and isotactic segments. The signal at 1145 cm^−1^ was regarded as a measure of polymer crystallinity. Additionally, there were peaks attributed to the vibration of O–H and C–O at 1088 and 1323 cm^−1^ characteristic for secondary alcohols. The bands at 448 and 608 cm^−1^ were associated with the deformation of C–C–C segments of polymer backbone [41]. Moreover, the two additional peaks at 1663 and 1000 cm^−1^ originated from the glutaraldehyde crosslinker and were attributed to the C=O and C–CH_2_ stretching vibrations, respectively.

The symmetric –COO^−^ stretching mode in Raman spectra of plain ALG membrane at 1434 cm^−1^ was an evidence of the interactions between alginate and crosslinking calcium ions, by observed shift from its typical position around 1450 cm^−1^ in a non-crosslinked matrix. The bands at 1607 and 1662 cm^−1^ were associated with the asymmetric stretching of –COO^−^ groups, while their deformation appeared at 1000 cm^−1^. Deformation modes of C–H bonds were attributed to the peaks at 1358 and 1307 cm^−1^. In turn, the group of signals at 1082, 959, 891, and 810 cm^−1^ originated from the combined vibration of asymmetric stretching of glycosidic linkage and symmetric stretching and deformation of C–O–C, C–C–H, and C–O–H units as well as skeletal stretching of C–C and C–O bonds. The bands close to 704 and 604 cm^−1^were assigned to the ring breathing and deformation, respectively. The deformation of C–C–C and C–O–C units was attributed to the bands at 446 and 399 cm^−1^[42].

The Raman bands of plain PVA and ALG membranes were sharper and of higher intensity as compared with corresponding signals present in the spectra of hybrid composite membranes. In case of hybrid membranes, the characteristic peaks of polymer matrix remained at the same positions and bands coming from the fillers, which had lower intensity, were not observed. PVA and ALG membranes filled with Fe(acac)_3_ were chosen as a representative composite membranes, for which the Raman spectra were presented (Figure 2).

#### 3.1.2. Swelling and Contact Angle Measurements

The results of swelling and contact angle measurements of plain and hybrid composite membranes are presented in Figure 3. A typical plain ALG membrane was found to have higher contact angle 38.21° and lower degree of swelling 197.82% than a PVA plain membrane (27.40° and 364.94%, respectively). Although both polymer matrix can be regarded as hydrophilic, the membranes fabricated from PVA were always more hydrophilic to a certain extent. This effect was less pronounced in case of hybrid composite membranes, especially loaded with Fe(acac)_3_. The lower hydrophilicity of ALG membranes was probably due to the presence of Ca^2+^ ions crosslinking this polymer. It is interesting that the introduction of glutaraldehyde crosslinker for PVA membranes, commonly regarded as hydrophobic, did not suppress the hydrophilicity of polymer matrix. Generally, the addition of iron-based fillers manifested through the suppression of membrane hydrophilicity—an increase in contact angle and a decrease in the degree of swelling were observed. Comparing the fillers, both iron oxides invoked similar effects on the hydrophilic properties of membrane, albeit membranes containing hematite were slightly more hydrophilic. A decrease in hydrophilicity was the most pronounced for Fe(acac)_3_. This organic iron complex has three acetic acid moieties in the structure, and is insoluble in water. Its less polar character makes him soluble only in organic solvents, which was reflected in the properties of membranes [43].

#### 3.1.3. Structural Analysis of Iron-base Fillers

The structure of synthesized iron oxides and Fe(acac)_3_ was confirmed by powder XRD measurements as shown in the Figure 4. Since a Cu K-alpha X-ray source was used, a background noise from fluoresced X-rays was increased.XRD pattern of synthesized hematite presented a peak system characteristic for a α-Fe_2_O_3_ with a rhombohedral phase (space group R-3c, ICCD Card No: 04-015-6947). The most prominent peaks matching the reference hematite phase were: 24.15° (012), 33.16° (104), 35.63° (110), 40.86° (113), 49.46° (024), 54.07° (116), 62.43° (214) and 64.0° (300). Based on the measured pattern, the sample structure was confirmed as a α-Fe_2_O_3_ and identified as the only phase. XRD pattern of magnetite was identified as the only phase in synthesized sample, and no other phases were identified in obtained diffractogram. All main peaks characteristic for magnetite with a face-centered cubic phase (space group Fd-3m, ICCD Card No: 00-011-0614) were present in the XRD spectrum (Figure 4). The most prominent peaks matching the magnetite phase were: 30.1° (220), 35.5° (311), 43.2° (400), 57.0° (511) and 62.6° (440). The structure of synthesised Fe(acac)_3_ was also in a good agreement with the reference data. The main peaks observed on the diffractogram matching a reference primitive orthorhombic phase were: 10.68° (020), 13.05° (consisting of two superimposed peaks, 13.03° (002) and 13.14° (201)), 16.87° (022), 21.33° (321), 22.76° (203), 23.73° (232) and 25.19° (223) (space group Pcab, ICCD Card No: 00-030-1763). XRD measurements confirmed that orthorhombic Fe(acac)_3_ was the only phase in a prepared sample. 

#### 3.1.4. Morphological Analysis

The cross-sectional morphology of the pristine and hybrid ALG and PVA membranes filled with hematite, magnetite and iron(III) acetylacetonate particles were observed using Phenom Pro-X SEM microscope equipped with an energy dispersive X-ray analyzer. The micrographs presented in Figure 5 showed that the cross-sectional view of a cryogenic fracture of plain ALG and PVA membranes was relatively smooth and no obvious interfacial defects were found. Still, both surfaces presented some fractographic features being the evidence of the fracture, which could indicate crack-pinning processes like small cusps, riverlines, or textured microflow. Interestingly, cross-sections of iron-loaded hybrid membranes based on PVA were found to provide a clean and flat final surface, especially when compared with iron-loaded ALG membranes. Although membranes exhibited some riverlines, the lack of debris, pits, and voids could indicate good adhesion between filler particles and the membrane material, and also their uniform distribution in the matrix. Consequently, PVA hybrid membrane was supposed to exhibit more brittle behavior than ALG, probably because of the extended, regular hydrogen bond between –OH groups. ALG matrix had longer polymer backbones and average molecular mass leading to its higher entanglement. The incorporation of fillers into both matrices was found to change only slightly the morphology of investigated hybrid membranes. In case of both iron oxides, the inconsiderable increase in micro-level roughness could be observed, but no evidence of aggregates formation and local cracks on the surface from chase separation could be found. Possible stacking and agglomeration of inorganic iron particles could have caused the formation of visible interphase cavities between the aggregates and polymer matrix. The iron(III) acetylacetonate filler was even better compatible with polymer matrix, what had the influence on the smoothness of a membrane.

### 3.2. Pervaporative Performance of Magnetic Hybrid Membranes

Fluxes and changes in concentration of ethanol acquired during the pervaporative dehydration of ethanol for both matrices and six different concentrations of three iron-based fillers became the basis for a comprehensive characteristic of transport processes.

Consequently, Figure 6 and Figure 7 show the evaluated parameters describing the effectiveness of pervaporative ethanol dehydration through investigated membranes. For all membranes, the flux increased significantly and continuously with the increase in filler content (Figure 6a,b). For both matrices, the highest flux was obtained for membranes filled with magnetite, and the lowest—for the membranes filled with hematite. The strong magnetic properties of ferromagnetic magnetite and magnetic field developed inside membrane were found to accelerate the permeation of water molecules. An increase in a flux could be also associated with the formation of “extra” free volume in polymer matrix in the presence of filler promoting the penetration of permeating particles through the membranes. It is known that for composite membranes, the nanofillers modify the packing of polymer chains. This phenomenon results in an enlargement of a free volume and an enhancement of measured transmembrane flux in comparison with a neat polymer [44]. The aggregation of the filler particles and their poor compatibility with polymer matrix results very likely in the creation of voids at the filler–matrix interface, which also change the free volume of hybrid composite membranes [45,46].

As it, a better compatibility between polymer matrix and paramagnetic organic complex Fe(acac)_3_ may have positive influence on the flux. Comparing both types of matrices, it can be seen that plain membranes have similar flux, but the introduction of any iron-based filler increases the flux of PVA membranes are ca. twice as that determined for ALG membranes (Figure 6c). Also Kuila et al. [47]. and Kulkarni et al. [48] observed this in their research, and associated mainly with differences in hydrophilic properties of matrices. The orientation of particles of magnetic fillers and their location between polymer chains modified the arrangement of polymer backbone and induced an increase in PVA crystallinity. Sabarudin et al. found that PVA crosslinked with glutaraldehyde provoked smaller and more homogenous crystallite size, indicating well-built lattice. For this reason, the magnetic susceptibility of Fe_3_O_4_ increased, and resulted in a facilitated permeation of water molecules through polymer matrix [49].

According to a solubility–diffusion model, both diffusion and solubility processes impact the permeation of water and ethanol molecules through membranes. The evolution of diffusion coefficients, solubility and permeation coefficients with increasing filler content for ALG and PVA membranes is presented in Appendix A, respectively. Because of the hydrophilic nature of both polymer matrices, all estimated coefficients are higher for water, which preferably permeates through membrane. The highest values of water diffusion coefficient are obtained for ALG membranes filled with Fe(acac)_3_ particles, however, the diffusion coefficient of ethanol is also relatively high in case of this particular membrane. On the other hand, for ALG membranes containing both iron oxide fillers, the diffusion coefficients of water remain unchanged or slightly decreased with the increase in filler content, and the diffusion coefficients of ethanol are relatively small and practically constant. In more hydrophilic PVA membranes, the highest diffusion coefficient of water is observed for a magnetite filled membrane. The high content of hematite is not advantageous in investigated membranes. For ALG matrix, because of a decreasing tendency of water diffusion coefficient, the difference between diffusion of water and ethanol becomes small, whereas in PVA membranes, starting from 10 wt% of hematite, both water and ethanol diffusion coefficients are practically the same. The solubility coefficient of ALG membranes filled with Fe(acac)_3_ is the smallest and nearly constant in a whole investigated range of filler content, while the highest water solubility coefficients are found for membranes loaded with hematite. In pursuance of Equation (6), the values of permeation coefficients are a product of diffusion and solubility coefficients. For low filled membranes, the evaluated values of permeation coefficient are quite similar. After addition of a filler, water permeation coefficient increases. For high filler content, water permeation is the highest for ALG membranes filled with magnetite. On the other hand, the addition of any filler does not have a significant effect on the ethanol permeation coefficient. The properties of matrices have the highest impact on permeation of ethanol molecules. Only for PVA membranes filled with the highest amount of hematite, i.e., 10 and 15 wt%, an increase in ethanol permeation coefficient is noticed, what is associated with an increase in the diffusion coefficient of ethanol mentioned for these two membranes earlier. It can be noticed that diffusion plays a dominant role in the transport of water through PVA membranes filled with Fe_3_O_4_, while solubility in case of Fe_2_O_3_ oxide, whereas for Fe(acac)_3_ both mechanisms have a similar share. Nevertheless, the diffusion plays a key role for ALG membranes loaded with iron(III) acetylacetonate particles, and solubility if hematite and magnetite are used as a filler.

Practical effectiveness expressed through separation factor and pervaporation separation index is shown in Figure 7A–D. For both types of polymer matrices, the separation effectiveness increases in the following order: Antiferromagnetic hematite < ferromagnetic magnetite < paramagnetic iron(III) acetylacetonate complex. The relation between separation effectiveness and filler concentration goes through a maximum in the 4–10 wt% range of filler concentration. The presence of the maximum amount of filler concentration, at which the highest selectivity of the membrane is obtained, is associated with a different aggregation process of filler particles inside polymer matrix. Germanos et al. [50] were found that the filling particles can move locally, thus creating spaces through which it is possible the diffusing of water molecules. As the concentration of the filler increases, the average distance between the particles shorten, resulting in a reduction in the crystalline structure of polymer matrix, as well as the formation of larger aggregations and thus spaces between them. Through such formed pathway, both water and ethanol molecules can move in unselective manner. As a consequence, the overall selectivity of membrane is diminished. [51]

Although the trend is similar for both matrices, the separation effectiveness of ALG membranes is as twice as the one corresponding with PVA membranes. The best value of α and *PSI* are found for ALG membranes containing 7 wt% of Fe(acac)_3_ (19.69 and ~16,000 *g·m^−2^·h^−1^*, respectively). A possible explanation is a spin pinning effect, which originates from Fe–O–C bonds formed between surface Fe atoms and carboxylic groups of ALG matrix [52]. Our previous results showed that magnetic properties of filler could highly enhance the efficiency of ethanol dehydration process [53]. Results achieved for Fe(acac)_3_ filler, presented in the current work, suggest that a better compatibility between organic filler and polymer matrix is even more important. Inorganic fillers are not permeable, so they decrease the active cross-section and obstruct the permeation. It seems that a synergistic effect between magnetic properties and organic nature of filler will take future advantage of better efficiency. As a potential candidate for such fillers, we propose molecular magnets that are widely employed in spintronics, but have not been used in membrane science so far.

Combining the values of flux and separation factor, that express the *PSI*, it can be noticed that despite a difference in *PSI* of plain ALG and PVA membranes, hybrid membranes show almost identical values of this parameter because the higher separation factor of ALG membranes is compensated by higher flux of membranes with PVA matrix.

## 4. Conclusions

In this paper, crosslinked hybrid PVA and ALG membranes loaded with hematite, magnetite, and iron(III) acetylacetonate as a filler are investigated in the dehydration of ethanol by pervaporation. The results of swelling and contact angle experiments show that the addition of a variety of iron fillers reduces the hydrophilicity of the polymer matrix. Comparing iron(III) acetylacetonate, hematite, and magnetite particles, it can be seen that the most hydrophilic membrane is the one containing hematite and the less—Fe(acac)_3_. SEM micrographs show that a cross-section of plain ALG and PVA membranes is relatively smooth and no obvious interfacial defects are found. Taking into account the structure of the membrane, the kind of polymer matrix and the dispersion of the filler, the difference in separation performance for individual fillers in ALG and PVA matrix can be observed. The best separation properties are found for the membrane filled with organic iron(III) acetylacetonate. Comparing both types of matrices, it can be seen that ALG membranes are characterized by a higher separation factor, whereas PVA membranes by a higher flux. Considering the separate ion properties of both investigated polymer matrix, regarding to *PSI*, both hybrid PVA and ALG are suitable for ethanol dehydration and the best results are achieved for membranes filled with Fe(acac)_3_. In this case, the highest *PSI* for ALG_Fe(acac)_3_ and PVA_Fe(acac)_3_ membranes are equal to 15,998 and 14,878 *g·m^−2^·h^−1^*, respectively.

Summarizing the obtained results, it can be concluded that the best efficiency in ethanol dehydration is achieved with membranes filled with organic compounds with defined magnetic properties. It seems to be prospective to synthesize of such compounds and use them as fillers in separation processes. Promising group such compounds and unexplored in membrane separation fields are molecular magnets recently developed for spintronics application.

## Figures and Tables

**Figure 1 materials-13-04152-f001:**
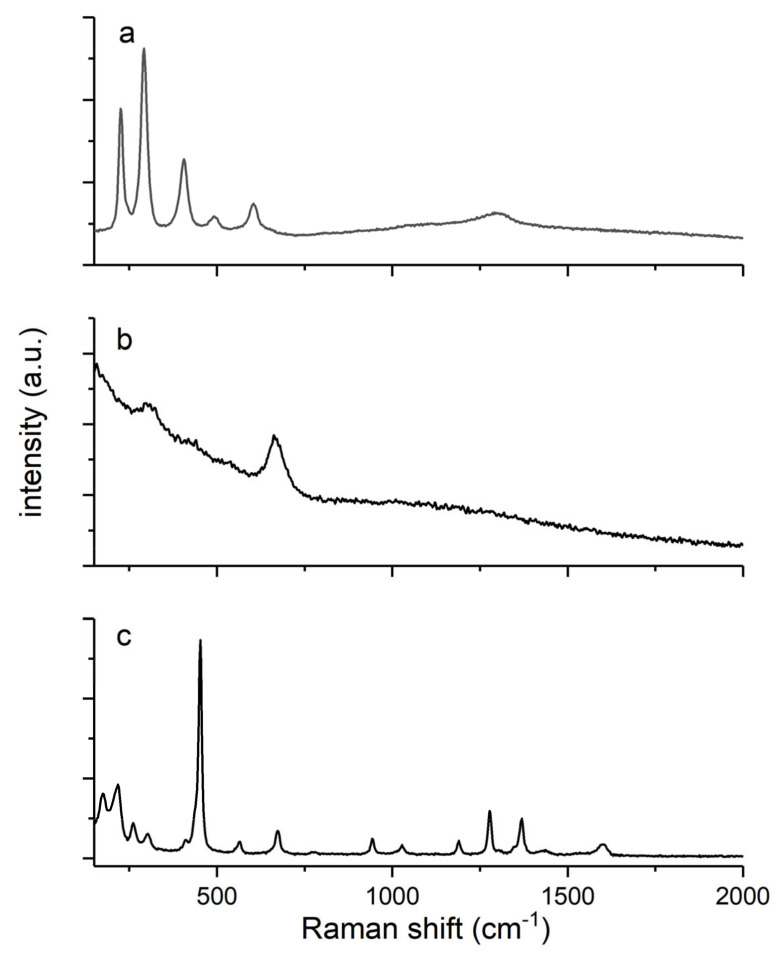
Raman spectra of synthesized iron-based filler: (**a**) hematite; (**b**) magnetite and (**c**) tris(acetylacetonato) iron(III).

**Figure 2 materials-13-04152-f002:**
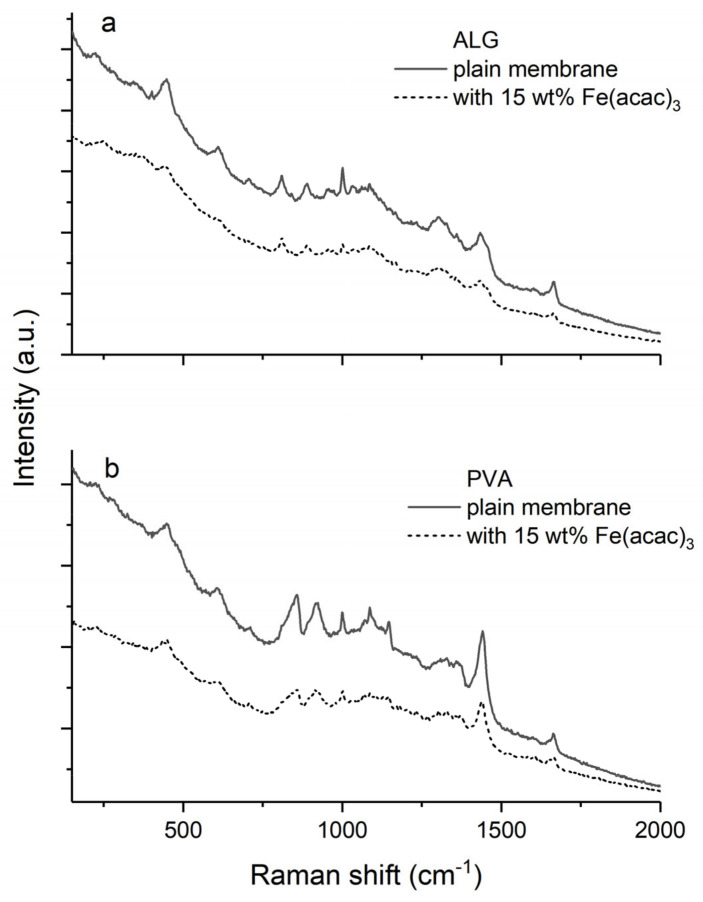
Raman spectra of plain and loaded with 15 wt% of tris(acetylacetonato) iron(III) membranes: (**a**) plain ALG membrane (solid line) and hybrid ALG membrane filled with Fe(acac)_3_ (dotted line); (**b**) plain PVA membrane (solid line) and hybrid PVA membrane filled with Fe(acac)_3_ (dotted line).

**Figure 3 materials-13-04152-f003:**
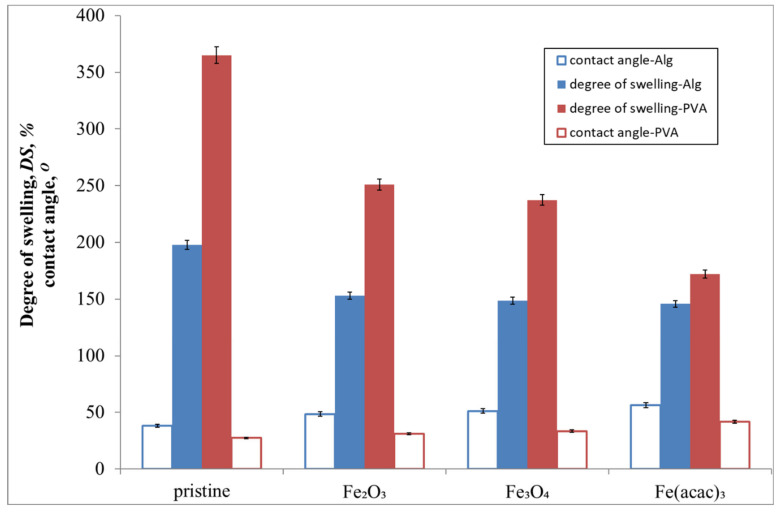
Average contact angles θ (n = 10) and degree of swelling DS (n = 10) of plain and hybrid composite PVA and ALG membranes filled with 15 wt% of different iron-based fillers.

**Figure 4 materials-13-04152-f004:**
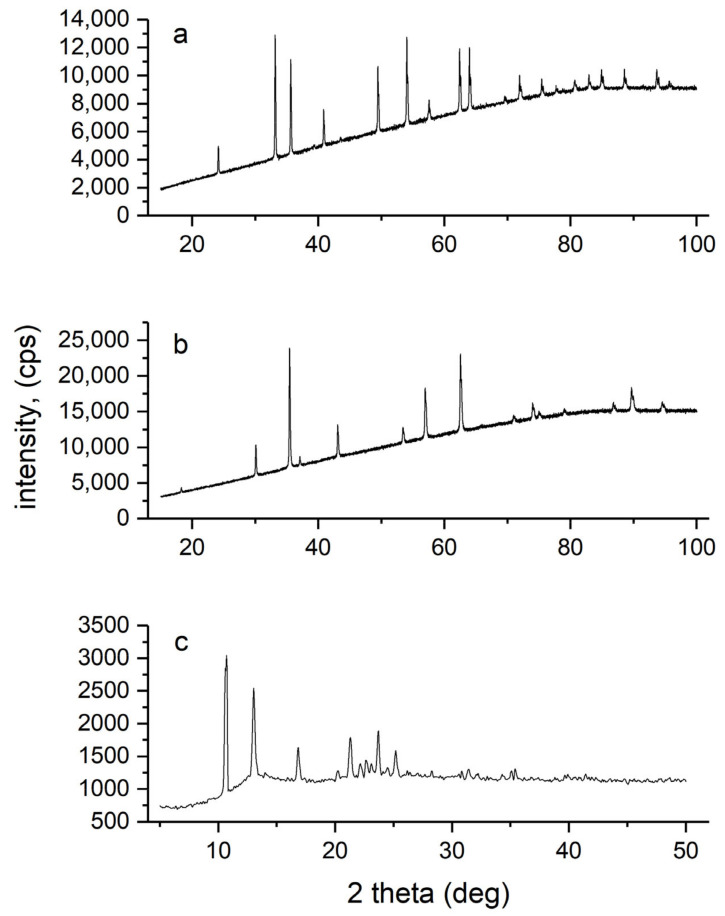
Powder X-rays diffractograms of synthesized iron-based fillers: (**a**) hematite; (**b** )magnetite and (**c**) tris(acetylacetonato) iron(III).

**Figure 5 materials-13-04152-f005:**
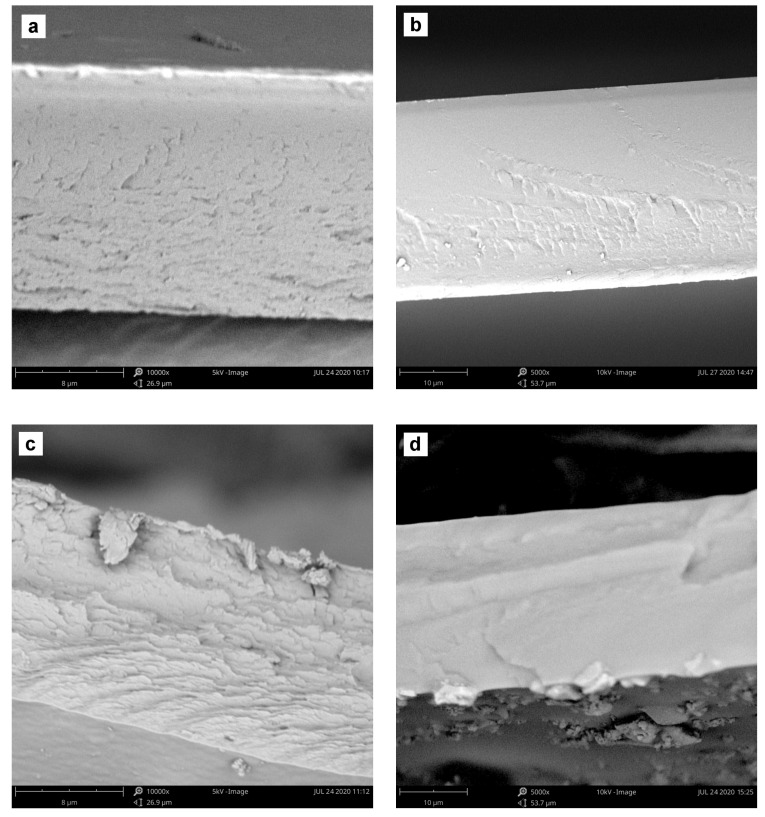
SEM images of the cross-sectional view of ALG (left column) and PVA (right column) membranes filled with 15 wt% of hematite, magnetite or iron(III) acetylacetonate, respectively. Samples were coated with 5 nm layer of Au. Magnification and acceleration voltage: 10,000× and 5 kV (ALG), and 5000× and 10 kV (PVA). (**a**) plain ALG; (**b**) plain PVA; (**c**) ALG_Fe_2_O_3_; (**d**) PVA_Fe_2_O_3_; (**e**) ALG_Fe_3_O_4_; (**f**) PVA_Fe_3_O_4_; (**g**) ALG_Fe(acac)_3_ and (**h**) PVA_Fe(acac)_3._

**Figure 6 materials-13-04152-f006:**
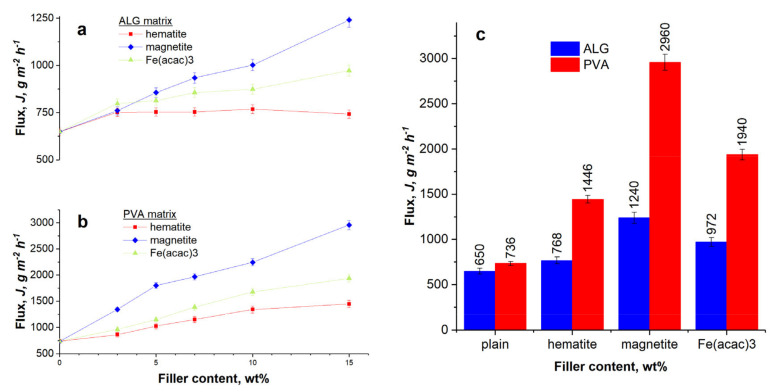
Dependence of the flux on a filler content for (**a**) ALG and (**b**) PVA membranes (symbols represent the experimental values and lines connect them only for visual guide), and (**c**) the comparison of the maximum achieved fluxes for both type of polymer matrices.

**Figure 7 materials-13-04152-f007:**
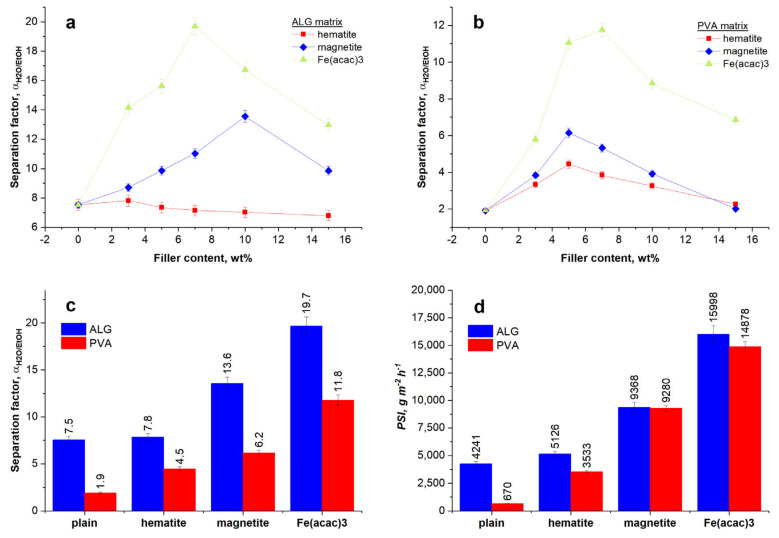
Changes of separation factor, $\alpha_{H_{2}O/EtOH}$ with a change in a filler content for (**a**) ALG and (**b**) PVA membranes (symbols represent the experimental values and lines connect them only for visual guide); and the comparison of the maximum values of (**c**) $\alpha_{H_{2}O/EtOH}$ and (**d**) *PSI* for both types of polymer matrices.

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
