# Peer review of "Collation Efficiency of Poly(Vinyl Alcohol) and Alginate Membranes with Iron-Based Magnetic Organic/Inorganic Fillers in Pervaporative Dehydration of Ethanol"

_materials, 2020, doi:10.3390/ma13184152_

Round 1

Reviewer 1 Report

The manuscript entitled:Collation efficiency of poly(vinyl alcohol) and alginate membranes with iron-based magnetic organic/inorganic fillers in pervaporative dehydration of ethanol , presents good insights in the field of hydrophilic pervaporation. The authors have enough characterized the obtained mixed matrix membranes and tested for the dehydration of ethanol. From my point of view, the article can be accepted after attending some minor corrections:

1)Introduction: The authors should give more examples of uses of pervaporations, e.g. assisting chemical and biochemical reactions (https://doi.org/10.1080/07388551.2019.1631248) and recovery of aromas from food systems (https://doi.org/10.1016/j.tifs.2019.12.003), production of non-alcoholic beverages, among others. The authors may check and cite the given literature reports.

2)Results: Through the characterization, the authors have found an enhancement of swelling degree of the pristine membranes by incorporating the organic/inorganic filler, which is attributted to the suppresion of polymer motion. Moreover, the membranes did not show any significant changes in contact angle measurements, but still show an enhanced separation factors. It can be seen that the permeation rates increased as a function of filler loading, but the authors have an separation factor enhancement until optimized point (Fig.7). Does the authors can give an explanation about it? Any presence unselective void/pathways? Please check the possible deffects created in mixed matrix membranes and their possible explanation of such phenomena, the authors should go further in the analysis of results.

3) Conclusion: More than giving the concluding remarks of their study, The authors should give a future reccomendations to the new authors in the field

Author Response

Answer to the Reviewer 1

  1. „Introduction: The authors should give more examples of uses of pervaporations, e.g. assisting chemical and biochemical reactions, (https://doi.org/10.1080/07388551.2019.1631248) and recovery of aromas from food systems (https://doi.org/10.1016/j.tifs.2019.12.003), production of non-alcoholic beverages, among others. The authors May check and cite the given literature reports”.

According to the Reviewer suggestion, the Introduction was rewritten and examples of another application of pervaporation processes were provided. The following sentence were added: „Furthermore, the pervaporation process has also found successful application in assisting chemical and biochemical reactions [7],  recovery of aromatic compounds from the food systems [8], and production of non-alcoholic beverages [9].”

  1. „Results: Through the characterization, the authors have found an enhancement of swelling degree of the pristine membranes by incorporating the organic/inorganic filler, which is attributed to the suppression of polymer motion. Moreover, the membranes did not show any significant changes in contact angle measurements, but still show an enhanced separation factors. It can be seen that the permeation rates increased as a function of filler loading, but the authors have an separation factor enhancement until optimized point (Fig.7). Does the authors can give an explanation about it? Any presence unselective void/pathways? Please check the possible defects created in mixed matrix membranes and their possible explanation of such phenomena, the authors should go further in the analysis of results”.

According to the Reviewer suggestion we changed the discussion of results in the following way:

The presence of the maximum amount of filler concentration, at which the highest selectivity of the membrane is obtained, is associated with a different aggregation process of filler particles inside polymer matrix. Germanos et al. [G.Germanos, S.Youssef, W.Farah, B.Lescop, S.Rioual, M.Abboud, The impact of magnetite nanoparticles on the physicochemical and adsorption properties of magnetic alginate beads. J. Environ. Chem. Eng., 8 (2020) 104223.] were found that the filling particles can move locally, thus creating spaces through which it is possible the diffusing of water molecules. As the concentration of the filler increases, the average distance between the particles shorten, resulting in a reduction in the crystalline structure of polymer matrix, as well as the formation of larger aggregations and thus spaces between them. Through such formed pathway, both water and ethanol molecules can move in unselective manner. As a consequence, the overall selectivity of membrane is diminished. [M. Nidhin, K.J. Sreeram, B.U. Nair, Polysaccharide films as templates in the synthesis of hematite nanostructures with special properties. Appl. Surf. Sci., 258 (2012) 5179–5184.]”.

  1. „Conclusion: More than giving the concluding remarks of their study, The authors should give a future recommendations to the new authors in the field”.

According to the Reviewer suggestion we added to the Conclusion the part about challenges and possible directions for further research related to the use of magnetic iron-filled membranes as following:

“Summarizing the obtained results, it can be concluded that the best efficiency in ethanol dehydration is achieved with membranes filled with organic compounds with defined magnetic properties. It seems to be prospective to synthesize of such compounds and use them as fillers in separation processes. Promising group such compounds and unexplored in membrane separation fields are molecular magnets recently developed for spintronics application”.

Reviewer 2 Report

The authors present 6 different mixed matrix membranes with three fillers, Fe2O3, Fe3O4 and Fe(acac)3 and two polymer matrix ALG and PVA for pervaporation. The paper also includes the characterizations including Ramen, XRD, SEM, etc. The paper is well organized and could be published with answering following questions. 

  1. Can author explain how to confirm the homogenous of the MMMs, maybe you can consider the EDS.
  2. The figures needs to include the error bar for those with contact angles Flux and selectivities. 
  3. How long is the pervaporation testing, is the MMMs stale during the test?

Author Response

Answer to the Reviewer 2

  1. „Can author explain how to confirm the homogenous of the MMMs, maybe you can consider the EDS".

EDS analysis could give some qualitative information but, unfortunately the resolution of such analyzer is too low to allow the precise evaluation of filer distribution. In the best way, the homogeneity of hybrid membranes can be checked by analyzing high resolution images from a TEM and/or SEM, e.g. by analyzing the membrane cross-sections it can be possible to asset the filler distribution. In case of poor compatibility, phase separation can be clearly seen. Another method is the use of fractal analysis supported image analysis described in the paper [G. Dudek, M. Krasowska, R. Turczyn, M. Gnus, A. Strzelewicz, Structure, morphology and separation efficiency of hybrid Alg/Fe3O4 membranes in pervaporative dehydration of ethanol. Separation and Purification Technology, 182 (2017) Pages 101-109]. By analyzing the fractal dimension and ΔD it is possible determine the self-similarity structure and, in consequence, distinguish the homogeneity of membranes.

  1. „The figures needs to include the error bar for those with contact angles, Flux and selectivities”. 

According to the Reviewer suggestion the errors bar were added to the Figures

  1. „How long is the pervaporation testing, is the MMMs stale during the test?”.

A single measurement takes ten hours. However, the measurements were repeated at least three times using the same membrane and no noticeable deviation was observed. The membranes were stable and did not show signs of damage or deterioration. Our experience shows that the tested membranes are really robust. The longest tested membrane was a chitosan one, filled with magnetite. It was tested for two months, ten hours each day.

Reviewer 3 Report

A well-written and put together study. I have no major quibbles. Your experiment design is solid and your discussion is in line with the findings as displayed. The presentation of data is excellent. 

You do have several formatting issues (section numbering, inconsistent font etc.) that need to be resolved during the revision process. The figure captions are not the format in the template provided. Discussion section should be 3 and 1. References need to be in the correct format as stated in the template. 

On Page 2 you say "make the membrane separation process is not within easy.". This is poorly phrased and the meaning is lost. Please rephrase. 

Author Response

Answer to the Reviewer 3

We would like to thank the Reviewer for the effort put into checking the manuscript and for his valuable comments and apologize for the formal mistakes arisen during the submission of manuscript.

  1. „You do have several formatting issues (section numbering, inconsistent font etc.) that need to be resolved during the revision process. The figure captions are not the format in the template provided. Discussion section should be 3 and 1. References need to be in the correct format as stated in the template”. 

According to the Reviewer suggestion we correct the formal issues of our manuscript, i.e. the references, figure captions and section numbering.

  1. „On Page 2 you say "make the membrane separation process is not within easy." This is poorly phrased and the meaning is lost. Please rephrase”. 

According to the Reviewer suggestion we have rewritten the mentioned sentence in the following manner: „Problems in ethanol dehydration result from the similarity of water and ethanol molecules’ physical properties, especially polarity, making the membrane separation process challenging”.
